# Generative models for graph-based protein design

**John Ingraham, Vikas K. Garg, Regina Barzilay, Tommi Jaakkola**
Computer Science and Artificial Intelligence Lab, MIT
{ingraham, vgarg, regina, tommi}@csail.mit.edu

## Abstract

Engineered proteins offer the potential to solve many problems in biomedicine, energy, and materials science, but creating designs that succeed is difficult in practice. A significant aspect of this challenge is the complex coupling between protein sequence and 3D structure, with the task of finding a viable design often referred to as the *inverse protein folding problem*. In this work, we introduce a conditional generative model for protein sequences given 3D structures based on graph representations. Our approach efficiently captures the complex dependencies in proteins by focusing on those that are long-range in sequence but local in 3D space. This graph-based approach improves in both speed and reliability over conventional and other neural network-based approaches, and takes a step toward rapid and targeted biomolecular design with the aid of deep generative models.

## 1 Introduction

A central goal for computational protein design is to automate the invention of protein molecules with defined structural and functional properties. This field has seen tremendous progess in the past two decades [1], including the design of novel 3D folds [2], enzymes [3], and complexes [4]. Despite these successes, current approaches are often unreliable, requiring multiple rounds of trial-and-error in which initial designs often fail [5, 6]. Moreover, diagnosing the origin of this unreliability is difficult, as contemporary bottom-up approaches depend both on the accuracy of complex, composite *energy functions* for protein physics and also on the efficiency of *sampling algorithms* for jointly exploring the protein sequence and structure space.

Here, we explore an alternative, top-down framework for protein design that directly learns a conditional generative model for protein *sequences* given a specification of the target structure, which is represented as a *graph* over the residues (amino acids). Specifically, we augment the autoregressive self-attention of recent sequence models [7] with graph-based representations of 3D molecular structure. By composing multiple layers of this structured self-attention, our model can effectively capture higher-order, interaction-based dependencies between sequence and structure, in contrast to previous parameteric approaches [8, 9] that are limited to only the first-order effects.

A graph-structured sequence model offers several benefits, including favorable computational efficiency, inductive bias, and representational flexibility. We accomplish the first two by leveraging a well-evidenced finding in protein science, namely that long-range dependencies in sequence are generally short-range in 3D space [10–12]. By making the graph and self-attention similarly sparse and localized in 3D space, we achieve computational scaling that is linear in sequence length. Additionally, graph structured inputs offer representational flexibility, as they accomodate both coarse, 'flexible backbone' (connectivity and topology) as well as fine-grained (precise atom locations) descriptions of structure.

We demonstrate the merits of our approach via a detailed empirical study. Specifically, we evaluate our model's performance for *structural generalization* to sequences of protein 3D folds that are topologically distinct from those in the training set. For fixed-backbone sequence design, we find that

our model achieves considerably improved statistical performance over a prior neural-network based model and also that it achieves higher accuracy and efficiency than Rosetta `fixbb`, a state-the-art program for protein design.

The rest of the paper is organized as follows. We first position our contributions with respect to the prior work in Section 1.1. We provide details on our methods, including structure representation, in Section 2. We introduce our *Structured Transformer* model in Section 2.2. The details of our experiments are laid out in Section 3, and the corresponding results that elucidate the merits of our approach are presented in Section 4.

## 1.1   Related Work

**Generative models for protein sequence and structure**   A number of works have explored the use of generative models for protein engineering and design [13]. [8, 9, 14] have used neural network-based models for sequences given 3D structure, where the amino acids are modeled independently of one another. [15] introduced a generative model for protein sequences conditioned on a 1D, context-free grammar based specification of the fold topology. Multiple works [16, 17] have modeled the conditional distribution of single amino acids given surrounding structure and sequence context with convolutional neural networks. In contrast to these works, our model captures the joint distribution of the full protein sequence while grounding these dependencies in terms of long-range interactions arising from structure.

In parallel to the development of structure-based models, there has been considerable work on deep generative models for protein sequences in individual protein families [18–21]. While useful, these methods presume the availability of a large number of sequences from a particular family, which are unavailable in the case of designing novel proteins that diverge significantly from natural sequences.

Several groups have obtained promising results using unconditional protein language models [22–25] to learn sequence representations that transfer well to supervised tasks. While serving different purposes, we emphasize that one advantage of *conditional* generative modeling is to facilitate adaptation to specific (and potentially novel) parts of structure space. Language models trained on hundreds of millions of evolutionary sequences will still be 'semantically' bottlenecked by the thousands of 3D evolutionary folds that these sequences represent. We propose evaluating protein language models with *structure*-based splitting of sequence data, and begin to see how unconditional language models may struggle to assign high likelihoods to sequences from out-of-training folds.

In a complementary line of research, several deep and differentiable parameterizations of protein structure [26–29] have been recently proposed that could be used to generate, optimize, or validate 3D structures for input to sequence design.

**Protein design and interaction graphs**   For classical approaches to computational protein design, which are based on joint modeling of structure and sequence, we refer the reader to a review of both methods and accomplishments in [1]. Many of the major 'firsts' in protein design are due to Rosetta [30, 31], a leading framework for protein design. More recently, there have been successes with non-parametric approaches to protein design [32] which are based on finding substructural homologies between the target and diverse templates in large protein database. In this work, we focus on comparisons to Rosetta (Section 4.2), since it is based on a single parametric energy function for capturing the sequence-structure relationship.

**Self-Attention**   Our model extends the Transformer [33] to capture sparse, pairwise relational information between sequence elements. The dense variation of this problem was explored in [34] and [35]. As noted in those works, incorporating general pairwise information incurs $\mathcal{O}(N^2)$ memory (and computational) cost for sequences of length $N$, which can be highly limiting for training on GPUs. We circumvent this cost by instead restricting the self-attention to the sparsity of the input graph. Given this graph-structured self-attention, our model may also be reasonably cast in the framework of message-passing or graph neural networks [36, 37] (Section 4.1). Our approach is similar to Graph Attention Networks [38], but augmented with edge features and an autoregressive decoder.

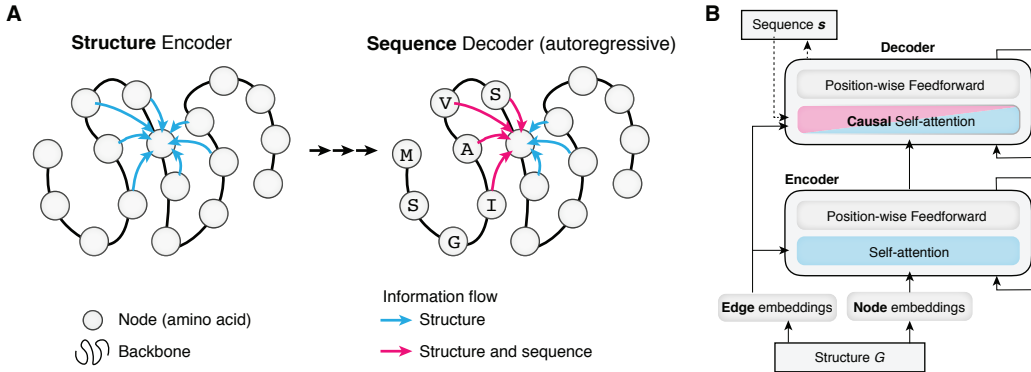

Figure 1: **A graph-based, autoregressive model for protein sequences given 3D structures.** (A) We cast protein design as language modeling conditioned on an input graph. In our architecture, an encoder develops a sequence-independent representation of 3D structure via multi-head self-attention [7] on the spatial $k$-nearest neighbors graph. A decoder then autoregressively generates each amino acid $s_i$ given the full structure and previously decoded amino acids. (B) Each layer of the encoder and decoder contains a step of neighborhood aggregation (self-attention) and of local information processing (position-wise feedforward).

## 2    Methods

In this work, we introduce a *Structured Transformer* model that draws inspiration from the self-attention based *Transformer* model [7] and is augmented for scalable incorporation of relational information (Figure 1). While general relational attention incurs quadratic memory and computation costs, we avert these by restricting the attention for each node $i$ to the set $\text{N}(i, k)$ of its $k$-nearest neighbors in 3D space. Since our architecture is multilayered, iterated local attention can derive progressively more global estimates of context for each node $i$. Second, unlike the standard Transformer, we also include edge features to embed the spatial and positional dependencies in deriving the attention. Thus, our model generalizes Transformer to spatially structured settings.

### 2.1    Representing structure as a graph

We represent protein structure in terms of an attributed graph $\mathcal{G} = (\mathcal{V}, \mathcal{E})$ with node features $\mathcal{V} = \{\boldsymbol{v}_1, \dots, \boldsymbol{v}_N\}$ describing each residue (amino acid, which are the letters which compose a protein sequence) and edge features $\mathcal{E} = \{\boldsymbol{e}_{ij}\}_{i \neq j}$ capturing relationships between them. This formulation can accommodate different variations on the macromolecular design problem, including both 'rigid backbone' design where the precise coordinates of backbone atoms are fixed, as well as 'flexible backbone' design where softer constraints such as blueprints of hydrogen-bonding connectivity [5] or 1D architectures [15] could define the structure of interest.

**3D considerations**    For a rigid-body design problem, the structure for conditioning is a fixed set of backbone coordinates $\mathcal{X} = \{\boldsymbol{x}_i \in \mathbb{R}^3 : 1 \leq i \leq N\}$, where $N$ is the number of positions[1]. We desire a graph representation of the coordinates $\mathcal{G}(\mathcal{X})$ that has two properties:

- *Invariance*. The features are invariant to rotations and translations.
- *Locally informative.* The edge features incident to $\boldsymbol{v}_i$ due to its neighbors $\text{N}(i, k)$, i.e. $\{\boldsymbol{e}_{ij}\}_{j \in \text{N}(i,k)}$, contain sufficient information to reconstruct all adjacent coordinates $\{\boldsymbol{x}_j\}_{j \in \text{N}(i,k)}$ up to rigid-body motion.

While invariance is motivated by standard symmetry considerations, the second property is motivated by limitations of current graph neural networks [36]. In these networks, updates to node features

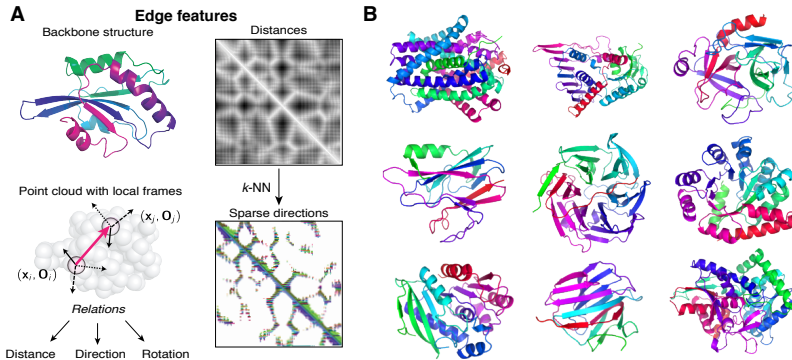

Figure 2: **Spatial features capture structural relationships across diverse folds.** (A) The edge features of our most detailed protein graph representation capture the relative distance, direction, and orientation between two positions on the backbone. For scalability, all computation after an initially dense Euclidean distance calculation (right, top), such as relative directions (right, bottom) and neural steps, can be restricted to the $k$-Nearest Neighbors graph. (B) Example of topological variation in the dataset. Protein chains in train, test, and validation are split by the sub-chain CATH [40] topologies, which means that folds in each set will have distinct patterns of spatial connectivity.

$v_i$ depend only on the edge and node features adjacent to $v_i$. However, typically, these features are insufficient to reconstruct the relative neighborhood positions $\{x_j\}_{j \in N(i,k)}$, so individual updates cannot fully depend on the 'local environment'. For example, when reasoning about the neighborhood around coordinate $x_i$, the pairwise distances $D_{ia}$ and $D_{ib}$ will be insufficient to determine if $x_a$ and $x_b$ are on the same or opposite sides.

**Relative spatial encodings**   We develop invariant and locally informative features by first augmenting the points $x_i$ with 'orientations' $O_i$ that define a local coordinate system at each point (Figure 2). We define these in terms of the backbone geometry as

$$O_i = [b_i \; n_i \; b_i \times n_i],$$

where $b_i$ is the negative bisector of angle between the rays $(x_{i-1} - x_i)$ and $(x_{i+1} - x_i)$, and $n_i$ is a unit vector normal to that plane. Formally, we have

$$u_i = \frac{x_i - x_{i-1}}{||x_i - x_{i-1}||}, \quad b_i = \frac{u_i - u_{i+1}}{||u_i - u_{i+1}||}, \quad n_i = \frac{u_i \times u_{i+1}}{||u_i \times u_{i+1}||}.$$

Finally, we derive the spatial edge features $e_{ij}^{(s)}$ from the rigid body transformation that relates reference frame $(x_i, O_i)$ to reference frame $(x_j, O_j)$. While this transformation has 6 degrees of freedom, we decompose it into features for distance, direction, and orientation as

$$e_{ij}^{(s)} = \left( r\left(||x_j - x_i||\right), \quad O_i^T \frac{x_j - x_i}{||x_j - x_i||}, \quad q\left(O_i^T O_j\right) \right).$$

Here the first vector is a *distance* encoding $r(\cdot)$ lifted into a radial basis[2], the second vector is a *direction* encoding that corresponds to the relative direction of $x_j$ in the reference frame of $(x_i, O_i)$, and the third vector is an *orientation* encoding $q(\cdot)$ of the quaternion representation of the spatial rotation matrix $O_i^T O_j$. Quaternions represent 3D rotations as four-element vectors that can be efficiently and reasonably compared by inner products [39].[3]

**Relative positional encodings**   As in the original Transformer, we also represent distances between residues in the sequence (rather than space) with positional embeddings $e_{ij}^{(p)}$. Specifically, we need to represent the positioning of each neighbor $j$ relative to the node under consideration $i$. Therefore,

we obtain the position embedding as a sinusoidal function of the gap $i - j$. We retain the sign of the distance $i - j$ because protein sequences are generally asymmetric. These relative encodings contrast the absolute encodings of the original Transformer, but are consistent with modifications described in subsequent work [34].

**Node and edge features**     Finally, we obtain an aggregate edge encoding vector $\boldsymbol{e}_{ij}$ by concatenating the structural encodings $\boldsymbol{e}_{ij}^{(s)}$ with the positional encodings $\boldsymbol{e}_{ij}^{(p)}$ and then linearly transforming them to have the same dimension as the model. We only include edges in the $k$-nearest neighbors graph of $\mathcal{X}$, with $k = 30$ for all experiments. This $k$ is generous, as typical definitions of residue-residue contacts in proteins will result in $< 20$ contacts per residue. For node features, we compute the three dihedral angles of the protein backbone $(\phi_i, \psi_i, \omega_i)$ and embed these on the 3-torus as $\{\sin, \cos\} \times (\phi_i, \psi_i, \omega_i)$.

**Flexible backbone features**     We also consider 'flexible backbone' descriptions of 3D structure based on topological binary edge features and coarse backbone geometry. We combine the relative positional encodings with two binary edge features: *contacts* that indicate when the distance between $C_\alpha$ residues at $i$ and $j$ are less than 8 Angstroms and *hydrogen bonds* which are directed and defined by the electrostatic model of DSSP [41]. For coarse node features, we compute virtual dihedral angles and bond angles between backbone $C_\alpha$ residues, interpret them as spherical coordinates, and represent them as points on the unit sphere.

## 2.2   Structured Transformer

**Autoregressive decomposition**     We decompose the distribution of a protein sequence given a 3D structure as

$$p(\boldsymbol{s}|\boldsymbol{x}) = \prod_i p(s_i|\boldsymbol{x}, \boldsymbol{s}_{<i}),$$

where the conditional probability $p(s_i|\boldsymbol{x}, \boldsymbol{s}_{<i})$ of amino acid $s_i$ at position $i$ is depends both on the input structure $\boldsymbol{x}$ as well as on the preceding amino acids $\boldsymbol{s}_{<i} = \{s_1, \dots s_{i-1}\}$.[4] These conditionals are parameterized in terms of two sub-networks: an *encoder* that computes refined node embeddings from structure-based node features $\mathcal{V}(\mathbf{x})$ and edge features $\mathcal{E}(\mathbf{x})$, and a *decoder* that autoregressively predicts letter $s_i$ given the preceding sequence and structural embeddings from the encoder (Figure 1).

**Encoder**     Our encoder module is designed as follows. A transformation $\boldsymbol{W}_h : \mathbb{R}^{d_v} \mapsto \mathbb{R}^d$ produces initial embeddings $\boldsymbol{h}_i = \boldsymbol{W}_h(\boldsymbol{v}_i)$ from the node features $\boldsymbol{v}_i$ pertaining to position $i \in [N] \triangleq \{1, 2, \dots, N\}$.

Each layer of the encoder implements a multi-head self-attention component, where head $\ell \in [L]$ can attend to a separate subspace of the embeddings via learned query, key and value transformations [7]. The queries are derived from the current embedding at node $i$ while the keys and values from the relational information $\boldsymbol{r}_{ij} = (\boldsymbol{h}_j, \boldsymbol{e}_{ij})$ at adjacent nodes $j \in N(i, k)$. Specifically, $\boldsymbol{W}_q^{(\ell)}$ maps $\boldsymbol{h}_i$ to *query* embeddings $\boldsymbol{q}_i^{(\ell)}$, $\boldsymbol{W}_z^{(\ell)}$ maps pairs $\boldsymbol{r}_{ij}$ to *key* embeddings $\boldsymbol{z}_{ij}^{(\ell)}$ for $j \in N(i, k)$, and $\boldsymbol{W}_v^{(\ell)}$ maps the same pairs $\boldsymbol{r}_{ij}$ to *value* embeddings $\boldsymbol{v}_{ij}^{(\ell)}$ for each $i \in [N], \ell \in [L]$. Decoupling the mappings for keys and values allows each to depend on different subspaces of the representation.

We compute the attention $a_{ij}^{(\ell)}$ between query $\boldsymbol{q}_i^{(\ell)}$ and key $\boldsymbol{z}_{ij}^{(\ell)}$ as a function of their scaled inner product:

$$a_{ij}^{(\ell)} = \frac{\exp(m_{ij}^{(\ell)})}{\sum_{j' \in N(i,k)} \exp(m_{ij'}^{(\ell)})}, \qquad \text{where} \quad m_{ij}^{(\ell)} = \frac{\boldsymbol{q}_i^{(\ell)\top} \boldsymbol{z}_{ij}^{(\ell)}}{\sqrt{d}}.$$

The results of each attention head $l$ are collected as the weighted sum

$$\boldsymbol{h}_i^{(\ell)} = \sum_{j \in N(i,k)} a_{ij}^{(\ell)} \boldsymbol{v}_{ij}^{(\ell)},$$

Table 1: **Null perplexities** for common statistical models of proteins.

| Null model | Perplexity | Conditioned on |
|---|---|---|
| Uniform | 20.00 | - |
| Natural frequencies | 17.83 | Random position in a natural protein |
| Pfam HMM profiles | 11.64 | Specific position in a specific protein family |

and then concatenated and transformed to give the update

$$\Delta \boldsymbol{h}_i = \boldsymbol{W}_o \, \text{Concat} \left( \boldsymbol{h}_i^{(1)}, \ldots, \boldsymbol{h}_i^{(L)} \right).$$

We update the embeddings with this residual, and alternate between these self-attention layers and position-wise feedforward layers as in the original Transformer [7]. We stack multiple layers atop each other, and thereby obtain continually refined embeddings as we traverse the layers bottom up. The encoder yields the embeddings produced by the topmost layer as its output.

**Decoder**   Our decoder module has the same structure as the encoder but with augmented relational information $\boldsymbol{r}_{ij}$ that allows access to the preceding sequence elements $\boldsymbol{s}_{<i}$ in a *causally consistent* manner. In contrast to the encoder, where the keys and values are based on the relational information $\boldsymbol{r}_{ij} = (\boldsymbol{h}_j, \boldsymbol{e}_{ij})$, the decoder can additionally access sequence elements $s_j$ as

$$\boldsymbol{r}_{ij}^{(\text{dec})} = \begin{cases} (\boldsymbol{h}_j^{(\text{dec})}, \boldsymbol{e}_{ij}, \mathbf{g}(s_j)) & i > j \\ (\boldsymbol{h}_j^{(\text{enc})}, \boldsymbol{e}_{ij}, \mathbf{0}) & i \le j \end{cases}.$$

Here $\boldsymbol{h}_j^{(\text{dec})}$ is the embedding of node $j$ in the current layer of the decoder, $\boldsymbol{h}_j^{(\text{enc})}$ is the embedding of node $j$ in the final layer of the encoder, and $\mathbf{g}(s_j)$ is a sequence embedding of amino acid $s_j$ at node $j$. This concatenation and masking structure ensures that sequence information only flows to position $i$ from positions $j < i$, but still allows position $i$ to attend to subsequent structural information unlike the standard Transformer decoder.

We now demonstrate the merits of our approach via a detailed empirical analysis. We begin with the experimental set up including our architecture, and description of the data used in our experiments.

## 3   Training

**Architecture**   In all experiments, we used three layers of self-attention and position-wise feedforward modules for the encoder and decoder with a hidden dimension of 128.

**Optimization**   We trained models using the learning rate schedule and initialization of the original Transformer paper [7], a dropout rate of $10\%$ [42], a label smoothing rate of $10\%$, and early stopping based on validation perplexity. The unconditional language models did not include dropout or label smoothing.

**Dataset**   To evaluate the ability of our models to generalize across different protein folds, we collected a dataset based on the CATH hierarchical classification of protein structure [40]. For all domains in the CATH 4.2 $40\%$ non-redundant set of proteins, we obtained full chains up to length 500 and then randomly assigned their CATH topology classifications (CAT codes) to train, validation and test sets at a targeted 80/10/10 split. Since each chain can contain multiple CAT codes, we first removed any redundant entries from train and then from validation. Finally, we removed any chains from the test set that had CAT overlap with train and removed chains from the validation set with CAT overlap to train or test. This resulted in a dataset of 18024 chains in the training set, 608 chains in the validation set, and 1120 chains in the test set. There is zero CAT overlap between these sets.

## 4   Results

A challenge in evaluating computational protein design methods is the *degeneracy* of the relationship between protein structure and sequence. Many protein sequences may reasonably design the same

Table 2: **Per-residue perplexities for protein language modeling** (lower is better). The protein chains have been cluster-split by CATH topology, such that test includes only unseen 3D folds. While a structure-conditioned language model can generalize in this structure-split setting, unconditional language models struggle.

| Test set | Short | Single chain | All |
|---|---|---|---|
| **Structure-conditioned models** | | | |
| Structured Transformer (ours) | **8.54** | **9.03** | **6.85** |
| SPIN2 [8] | 12.11 | 12.61 | - |
| **Language models** | | | |
| LSTM ($h = 128$) | 16.06 | 16.38 | 17.13 |
| LSTM ($h = 256$) | 16.08 | 16.37 | 17.12 |
| LSTM ($h = 512$) | 15.98 | 16.38 | 17.13 |
| Test set size | 94 | 103 | 1120 |

3D structure [43], meaning that sequence similarity need not necessarily be high. At the same time, single mutations may cause a protein to break or misfold, meaning that high sequence similarity isn't sufficient for a correct design. To deal with this, we will focus on three kinds of evaluation: (i) likelihood-based, where we test the ability of the generative model to give high likelihood to held out sequences, (ii) native sequence recovery, where we evaluate generated sequences vs the native sequences of templates, and (iii) experimental comparison, where we compare the likelihoods of the model to high-throughput data from a de novo protein design experiment.

We find that our model is able to attain considerably improved statistical performance in its likelihoods while simultaneously providing more accurate and efficient sequence recovery.

## 4.1 Statistical comparison to likelihood-based models

**Protein perplexities** What kind of perplexities might be useful? To provide context, we first present perplexities for some simple models of protein sequences in Table 1. The amino acid alphabet and its natural frequencies upper-bound perplexity at 20 and ~17.8, respectively. Random protein sequences under these null models are unlikely to be functional without further selection [44]. First order profiles of protein sequences such as those from the Pfam database [45], however, are widely used for protein engineering. We found the average perplexity per letter of profiles in Pfam 32 (ignoring alignment uncertainty) to be ~11.6. This suggests that even models with high perplexities high as ~ 11 have the potential to be useful for the space of functional protein sequences.

**The importance of structure** We found that there was a significant gap between unconditional language models of protein sequences and models conditioned on structure. Remarkably, for a range of structure-independent language models, the typical test perplexities are ~16-17 (Table 2), which were barely better than null letter frequencies (Table 1). We emphasize that the RNNs were not broken and could still learn the training set in these capacity ranges. All structure-based models had (unsurprisingly) considerably lower perplexities. In particular, our Structured Transformer model attained a perplexity of ~7 on the full test set. It seems that protein language models trained on one subset of 3D folds (in our cluster-splitting procedure) generalize poorly to predict the sequences

Table 3: **Ablation of graph features and model components**. Test perplexities (lower is better).

| Node features | Edge features | Aggregation | Short | Single chain | All |
|---|---|---|---|---|---|
| **Rigid backbone** | | | | | |
| Dihedrals | Distances, Orientations | Attention | 8.54 | 9.03 | 6.85 |
| Dihedrals | Distances, Orientations | PairMLP | **8.33** | **8.86** | **6.55** |
| $C_\alpha$ angles | Distances, Orientations | Attention | 9.16 | 9.37 | 7.83 |
| Dihedrals | Distances | Attention | 9.11 | 9.63 | 7.87 |
| **Flexible backbone** | | | | | |
| $C_\alpha$ angles | Contacts, Hydrogen bonds | Attention | 11.71 | 11.81 | 11.51 |

| Method | Recovery (%) | Speed (AA/s) CPU | Speed (AA/s) GPU |
|---|---|---|---|
| Rosetta 3.10 `fixbb` | 17.9 | $4.88 \times 10^{-1}$ | N/A |
| Ours ($T = 0.1$) | **27.6** | $\mathbf{2.22 \times 10^2}$ | $\mathbf{1.04 \times 10^4}$ |

(a) Single chain test set (103 proteins)

| Method | Recovery (%) |
|---|---|
| Rosetta, `fixbb` 1 | 33.1 |
| Rosetta, `fixbb` 2 | 38.4 |
| Ours ($T = 0.1$) | **39.2** |

(b) Ollikainen benchmark (40 proteins)

Table 4: **Improved reliability and speed compared to Rosetta.** (a) On the 'single chain' test set, our model more accurately recovers native sequences than Rosetta `fixbb` with greater speed (CPU: single core of Intel Xeon Gold 5115, GPU: NVIDIA RTX 2080). This set includes NMR-based structures for which Rosetta is known to not be robust [46]. (b) Our model also performs favorably on a prior benchmark of 40 proteins. All results reported as median of average over 100 designs.

of unseen folds. We believe this possibility might be important to consider when training protein language models for protein engineering and design.

**Improvement over deep profile-based methods**   We also compared to a recent method SPIN2 that predicts, using deep neural networks, protein sequence profiles given protein structures [8]. Since SPIN2 is computationally intensive (minutes per protein for small proteins) and was trained on complete proteins rather than chains, we evaluated it on two subsets of the full test set: a 'Small' subset of the test set containing chains up to length 100 and a 'Single chain' subset containing only those models where the single chain accounted for the entire protein record in the Protein Data Bank. Both subsets discarded any chains with structural gaps (chain break). We found that our Structured Transformer model significantly improved upon the perplexities of SPIN2 (Table 2).

**Graph representations and attention mechanisms**   The graph-based formulation of protein design can accommodate very different formulations of the problem depending on how structure is represented by a graph. We tested different approaches for representing the protein including both more 'rigid' design with precise geometric details, and 'flexible' topological design based on spatial contacts and hydrogen bonding (Table 3). For the best perplexities, we found that using local orientation information was indeed important above simple distance measures. At the same time, even the topological features were sufficient to obtain better perplexities than SPIN2 (Table 2), which uses precise atomic details.

In addition to varying the graph features, we also experimented with an alternative aggregation function from message passing neural networks [36].[5] We found that a simple aggregation function $\Delta \mathbf{h}_i = \sum_j \mathrm{MLP}(\boldsymbol{h}_j, \boldsymbol{h}_j, \boldsymbol{e}_{ij})$ led to the best performance of all models, where $\mathrm{MLP}(\cdot)$ is a two layer perceptron that preserves the hidden dimension of the model. We speculate that this is due to potential overfitting by the attention mechanism. Although this suggests room for future improvements, we use multi-head self-attention throughout the remaining experiments.

### 4.2   Benchmarking protein redesign

**Decoding strategies**   Generating protein sequence designs requires a sampling scheme for drawing high-likelihood sequences from the model. While beam-search or top-$k$ sampling [47] are commonly used heuristics for decoding, we found that simple biased sampling from the temperature adjusted distributions $p^{(T)}(\mathbf{s}|\boldsymbol{x}) = \prod_i \frac{p(s_i|\boldsymbol{x}, \boldsymbol{s}_{<i})^{1/T}}{\sum_a p(a|\boldsymbol{x}, \boldsymbol{s}_{<i})^{1/T}}$ was sufficient for obtaining sequences with higher likelihoods than native. We used a temperature of $T = 0.1$ selected from sequence recovery on validation. For conditional redesign of a subset of positions in a protein, we speculate that the likelihood calculation is sufficiently fast such that MCMC-based approaches such as Gibbs sampling may be feasible.

Table 5: **Structure-conditioned likelihoods correlate with mutation effects in *de novo*-designed miniproteins**. Shown are Pearson correlation coefficients ($R$, higher is better) between the log-likelihoods of mutated sequences and high-throughput mutation effect data from a systematic design of miniproteins [6]. Each design (column) includes 775 experimentally tested mutant protein sequences.

| Design | $\beta\beta\alpha\beta\beta_{37}$ | $\beta\beta\alpha\beta\beta_{1498}$ | $\beta\beta\alpha\beta\beta_{1702}$ | $\beta\beta\alpha\beta\beta_{1716}$ | $\alpha\beta\beta\alpha_{779}$ |
|---|---|---|---|---|---|
| **Rigid backbone** | 0.47 | 0.45 | 0.12 | 0.47 | 0.57 |
| **Flexible backbone** | 0.50 | 0.44 | 0.17 | 0.40 | 0.56 |

| Design | $\alpha\beta\beta\alpha_{223}$ | $\alpha\beta\beta\alpha_{726}$ | $\alpha\beta\beta\alpha_{872}$ | $\alpha\alpha\alpha_{134}$ | $\alpha\alpha\alpha_{138}$ |
|---|---|---|---|---|---|
| **Rigid backbone** | 0.36 | 0.11 | 0.21 | 0.24 | 0.33 |
| **Flexible backbone** | 0.33 | 0.21 | 0.23 | 0.36 | 0.41 |

**Comparison to Rosetta** To evaluate the performance of our model at generating realistic protein sequences, we performed two experiments that compare with Rosetta [30], a state-of-the-art framework for computational protein design. We found that our model was more accurate and significantly faster than Rosetta (Table 4). In the first, we used the latest version of Rosetta (3.10) to design sequences for our 'Single chain' test set with the *fixbb* fixed-backbone design protocol and default parameters (Table 4a). In the second, we also compared to a prior benchmark from members of the Rosetta community [48, 49] across 40 diverse proteins. For this set of proteins, we re-split our dataset to form new training and validation sets with no CAT overlap to the 40 templates for design. Although this reduced the size of the training set from ∼18,000 to ∼10,000 chains, we found our model to be both more accurate than and several orders of magnitude faster than Rosetta (Table 4b).

### 4.3 Unsupervised anomaly detection for experimental protein design

We can also measure what our structure-conditioned language model 'knows' about protein function by comparing the likelihoods it assigns to functional and non-functional mutant proteins from recent high-throughput design experiments. This may be seen as a kind of evolutionary *unsupervised anomaly detection* [18]. We compare to a recent high-throughput design and mutagenesis experiment in which several *de novo* designed mini-proteins were subject to systematic mutagenesis to all possible point mutations [6]. We find that the log-likelihoods of our model non-trivially reflect mutational preferences of designed proteins (Table 5). Importantly, we see that the performance is not dependent on precise 3D geometric features (e.g. distances and orientations) but can also be realized with coarse information (e.g. contacts, hydrogen bonds, and coarse backbone angles).

## 5 Conclusion

We introduced a new generative model for designing protein sequences based on a graph-representation of their 3D structures[6]. Our model augments the traditional sequence-level self-attention of Transformers [7] with graph-based, 3D structural encodings and is able to leverage the spatial locality of dependencies in molecular structures for efficient computation. When evaluated on unseen folds, our model achieves significantly improved perplexities over recent neural network-based generative models and generates sequences with improved accuracy and speed over a state-of-art program Rosetta.

Our framework suggests the possibility of being able to efficiently design and engineer protein sequences with structurally-guided deep generative models, and underscores the central role of modeling sparse, long-range dependencies in biological sequences.

### Acknowledgments

We thank members of the MIT MLPDS consortium, the MIT NLP group, and the reviewers for helpful feedback. This work was supported by the Machine Learning for Pharmaceutical Discovery and Synthesis (MLPDS) consortium.

## Footnotes

[1]Here we consider a single representative coordinate per position when deriving edge features but may revisit multiple atom types per position for features such as backbone angles or hydrogen bonds.

[2]We used 16 Gaussian RBFs isotropically spaced from 0 to 20 Angstroms.

[3]We represent quaternions in terms of their vector of real coefficients.

[4]We anticipate that alternative orderings for decoding the sequence may be favorable but leave this to future work.

[5]We thank one of our reviewers for this suggestion.

[6]Code is available at `github.com/jingraham/neurips19-graph-protein-design`.

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
