[Reviews · NeurIPS 2019]

Reviewer 1



I think the idea of using attention or transformer inspired architectures for protein modelling is useful and the authors changes on the standard transformer are helpful since the full attention computations are typically costly.

Reviewer 2



This paper addresses the problem of generation of protein sequences for a desired 3D structure, also known as the “inverse protein folding problem”. The authors introduce a model inspired by recent advances in language modeling (for the sequence decoder part of the model) and graph representation learning (for the encoder part of the model). Protein structures are represented as k-NN graphs, enriched with orientation/location-based features and features based on structural bindings. The encoder takes the form of an adapted Graph Attention Network, here termed “Structured Transformer”, which is enriched with edge features and relative positional encodings, and the decoder takes the form of an auto-regressive Transformer-based model. Results indicate improvements over a recent deep neural network baseline for this task. The problem is of high significance and the authors make several non-trivial contributions that clearly improve the state of the art in this area. The paper is very well written and generally of high quality. It is well-positioned w.r.t. related work and all the contributions are well-motivated. I am not an expert in the area of “inverse protein folding”, but the paper did a great job at introducing the problem and related work. It would be good, however, to provide a more detailed description of the SPIN2 baseline and discuss differences to this particular model. The experiments are chosen well, but it would be nice to see error bars on results and further ablation studies, especially on the proposed attention mechanism and the relative positional encodings. Overall, I can recommend this paper for acceptance. The authors seem to take it as a given that a Transformer-based model is naturally the best fit for this task, but I wonder whether a (potentially simpler) message passing neural network as in Gilmer et al. (ICML 2017) would perform similarly when used as an encoder for this task. To be more precise, this would correspond to performing a) an ablation study on the attention mechanism (i.e., leaving out a_{ij} in the update for h_i), and b) using a small MLP to transform v_{ij} instead of a linear map — this corresponds to the message in the message passing neural network framework. --- My questions have been addressed in the rebuttal and I am looking forward to seeing the comparison against Message Passing Neural Networks and a discussion of the SPIN baseline in the updated version of the paper. My recommendation remains accept, but I leave it to the other reviewers to judge the relevance of the new results comparing their method against non-deep learning baselines (for which I do not have any expertise).

Reviewer 3



Originality: Overall, the approach taken in this work differs significantly from past work. Unlike previous applications of deep learning to protein design, the authors present a model capable of modeling the joint distribution of protein sequences conditioned on their structure. The authors extend the Transformer to include sparse relations and handle data that is spatially structured. The authors provide a new representation of protein structure using a graph with relative spatial encodings. This representation is likely useful beyond the scope of protein design. Quality: The submission is technically sound and the experimental results are clearly described and analyzed. I do have one concern: in order for this algorithm to be practically useful, there must be a way to decode a sequence with high probability under the model given an input structure. By only looking at log-likelihoods of individual sequences in the evaluation metrics, the authors avoid using any decoding strategies. One concern is that given the large length of protein sequences, greedy decoding strategies such as beam-search will produce low quality sequences (i.e. sequences with low log-likelihoods). A simple experiment would be comparing log-likelihoods of the top sequences from the authors choice of a decoding strategy to the log-likelihood of native sequences. I would ask the authors to include a discussion of decoding strategies and experiments in the paper. Clarity: This paper is well written. The related works section is thorough and makes clear what the deficiencies of previous deep learning approaches are. The explanations of the graph representation and neural architecture are also clear. The results section provides a nice analysis of perplexities based on random protein sequences and first order profiles of protein sequences from the Pfam database. There were a few aspects of the paper that could use clarification and/or expansion. 1) I do not think the authors explain what the input backbone coordinates are? Since there is one coordinate for each residue, I would assume the authors are using the C_alpha coordinate. 2) I believe there is a typo on line 198. "We find that the model is able to attain considerably improved statistical performance over other statistical performance." Significance: The results provide a significant improvement over previous deep learning approaches but fail to make any comparison to traditional protein design algorithms. In my opinion, these comparisons would provide substantial value by illustrating how deep learning methods compare to conventional approaches. Nonetheless, the approach that the authors take is unique and seems to work reasonably well. The representations and architecture are likely useful outside of the scope of protein design. The authors do not provide any discussion of protein redesign, in which only a subset of the amino acid residues are designed and the rest are maintained at wild-type. The approach seems sub-optimal for redesign since it is tied to a particular choice of a decoding order. The only option I see to perform redesign with this method is to enforce the amino acids that are not being designed to their identities during the decoding. However, this might require more sophisticated decoding strategies to produce high-probability sequences. This limits the practical use of the method since full design is rarely the task at hand for practical protein design problems. --- My primary concerns were addressed in the rebuttal stage. I am updating my score from a (6) to a (7).

[Author Response · NeurIPS 2019]

**Summary** We are grateful to the reviewers for their thoughtful feedback, and for acknowledging the novelty of our approach and its potential impact for computational protein design. Based on their suggestions, we benchmark our method against additional non-deep-learning, state-of-the-art baselines. Our method achieves competitive accuracy with significantly accelerated and streamlined computation.

## Reviewer #1

**Comparing to state-of-art baselines** We have extended our experiments to include two benchmarks comparing against Rosetta, the leading framework for computational protein design. Following the suggestions of reviewer #4, we focus on 'native sequence recovery', which measures the model's ability to accurately recover sequences given a backbone structure alone. We evaluate native sequence recovery on two different training sets of proteins and find that our method is competitive on both (Table 1). In the first, we used the latest version of Rosetta (3.10) to design sequences specific to our test set with the *fixbb* fixed-backbone design protocol and default parameters (Table 1, left). In the second, we also compared to a prior benchmark from members of the Rosetta community (Kortemme group, PLOS one, 2015) across 40 diverse proteins (Table 1, right). To test our model against this, we re-split our dataset to form new training/validation sets that have no CATH topology overlap with their benchmark. This reduced the size of the training set from $\sim$18,000 chains to $\sim$10,000 chains, but we still found our model to be competitive with Rosetta.

We believe that achieving performance competitive with Rosetta (for this specific task) is a significant accomplishment, given it is built on several million lines of code developed by over 50 labs for two decades. We note that the Rosetta `fixbb` program emits an approximately 14,000-line usage message describing options if you add the flag `-help`.

| Method | Recovery (%) | Speed (residues/s) |
|---|---|---|
| Rosetta 3.10 `fixbb` | 17.9 | $4.88 \times 10^{-1}$ |
| Ours ($T = 0.1$) | **28.5** | $\mathbf{1.08 \times 10^4}$ |

(a) Single chain test set

| Method | Recovery (%) |
|---|---|
| Rosetta, `fixbb` 1 | 33.1 |
| Rosetta, `fixbb` 2 | 38.4 |
| Ours ($T = 0.1$) | **38.6** |

(b) Ollikainen 40 benchmark

Table 1: **Evaluation against Rosetta for native sequence recovery** (Left) Our model more accurately recovers native sequences than Rosetta `fixbb` (median sequence similarity to native across 111 structures, 100 designs per structure). We note that these numbers are generally low because our test set is enriched for difficult examples that come from NMR-based templates. (Right) Evaluation with a prior benchmark of 40 structures, 100 designs per structure.

## Reviewer #3

**Error bars and attention ablations** Thank you for these great suggestions. We agree that the presented framework might also be realized with message passing neural networks and that it would be interesting to understand the tradeoffs of non-attentive aggregation and other message nonlinearities. While we were not able to report those results at this point, we will include them in the camera-ready if accepted.

**Explanation of SPIN2** Thank you for this suggestion. We will expand on our discussion of methods behind baselines (including Rosetta). Briefly, SPIN2 uses a neural network based on local molecular environment features (local angles, contacts, fragment profiles) to predict the identity of that specific amino acid (rather than the joint like ours).

## Reviewer #4

**Decoding strategies** We found that we could generate sequences with considerably higher likelihoods than native state simply via biased sampling with a softmax temperature $T < 1$ (Figure 1). We agree that this is an important consideration and will add both the experiment and disscussion of strategies (such as beam search and top-$k$ sampling) in the paper.

**Benchmarking against Rosetta** Please see our response to reviewer # 1.

**Redesign** We will discuss two methods for redesign: first, because the likelihood calculation is reasonably fast (16,000 residues / s on a consumer GPU), the log-likelihood could be used out-of-the-box for MCMC-based sampling (e.g. Gibbs). Second, the model could be retrained on randomized permutations at training time (Uria et al, ICML, 2014), and then conditioned at test time on autoregressive orderings that put the designed residues last.

Figure 1: **Decoding.** Sampling with a low-temperature softmax (x-axis) generates sequences with higher normalized log likelihoods (y-axis) than native (horizontal line).

[Meta-Review · NeurIPS 2019]

There is a consensus among reviewers that this paper deals with a high impact problem, makes progress on solving it as compared to state-of-the-art, and is well written.